# Introducing an Organizational Perspective in SDG Implementation in the Public Sector in Spain: The Case of the Former Ministry of Agriculture, Fisheries, Food and Environment

**Miguel Soberón** [1,*] **, Teresa Sánchez-Chaparro** [1,2] **, Julia Urquijo** [1,3] **and David Pereira** [1,3]

1    Centro de Innovación en Tecnología para el Desarrollo Humano, Universidad Politécnica de
     Madrid (itdUPM), 28040 Madrid, Spain; teresa.sanchez@upm.es (T.S.-C.); julia.urquijo@upm.es (J.U.);
     d.pereira@upm.es (D.P.)
2    Department of Organizational Engineering, Business Administration and Statistics,
     Escuela Técnica Superior de Ingenieros Industriales, Universidad Politécnica de Madrid,
     28006 Madrid, Spain
3    Department of Agroforestry Engineering, Escuela Técnica Superior de Ingeniería Agronómica y de
     Biosistemas (ETSIAAB), Universidad Politécnica de Madrid, 28040 Madrid, Spain
*    Correspondence: miguel.soberon@upm.es; Tel.: +34-660-086-327

**Abstract:** The public sector has an indisputable role in the implementation of the Sustainable Development Goals (SDGs). However, the interrelated nature of the SDGs represents a challenge for the public sector, which has in the last few decades undergone a process of specialization, decentralization and fragmentation. Hence, the establishment of coordination mechanisms within the public sector are needed to ensure implementation. This article introduces an organizational perspective in a participative SDG prioritization process carried out by a public organization: the former Spanish Ministry of Agriculture, Fisheries, Food and Environment (MAPAMA). A case study methodology is used to identify internal collaboration needs in order to address the SDGs and to analyze the driving and restraining forces operating within the organization so that the required organizational changes can be initiated. Our findings reveal that the organizational perspective is key in supporting SDG implementation and boosting the transformative capacity that underpins the 2030 Agenda. Public organizations must combine different coordination approaches, according to the demands that each specific SDG target makes upon the organization. Furthermore, engaging internal agents in participative processes for the development of the implementation is essential to reproducing the dynamics of internal collaboration that will be needed in future stages of the adoption of the 2030 Agenda.

**Keywords:** sustainable development goals (SDGs); public sector; organizational perspective; internal collaboration; prioritization process

## 1. Introduction

The public sector is a key actor for the successful implementation of the Sustainable Development Goals (SDGs) of the 2030 Agenda [1–4]. In the first place, having a sound public administrative system is a stand-alone objective, as SDG 16 aims at building "effective, accountable and inclusive institutions at all levels". Secondly, the public sector is specifically called upon to implement different policy instruments so as to ensure wide access to public services; adopt policies and strategies to achieve certain SDGs such as gender equality (SDG 5) and job creation and entrepreneurship (SDG 8); and invest

resources in different instrumental areas, such as in research and innovation (SDG 9) and multisector partnerships (SDG 17). Finally, the whole agenda stands upon the idea of active coordination and partnerships among the public, private and nonprofit sectors for the realization of the goals [2–6].

A particular challenge in the adoption of the 2030 Agenda by the public sector is to find ways for public organizations to engage in SDGs by establishing and implementing an appropriate sustainability strategy [7,8]. Although there is no unique way of supporting this process [9], prioritization is recognized in academic and practitioner literature as a crucial initial step, as it enables focusing on a reduced set of priorities [7], thus making SDG implementation more effective and manageable [9]. Priorities should be established through coherent and structured approaches [7,10], as otherwise there is a risk of "cherry picking" the goals that the organization was already working on and not dealing with those that were left out but were still important for the organization or its stakeholders [11], which can lead to reinforcing "business as usual" and stopping the transformational character of the 2030 Agenda [12,13].

Along with establishing a set of priorities, important structural and behavioral changes are likely to be required at the organizational level to address sustainability challenges [14,15]. As the SDGs are highly interrelated, progress towards one SDG target is also linked to other SDG targets through complex feedback, creating important requirements in terms of intraorganizational and interorganizational coordination and collaboration [7,12,16–20]. Although this is true for any kind of organization, this statement is particularly relevant in the case of the public sector, which has been affected by a trend in the last decades towards specialization, decentralization and fragmentation, and which has ultimately resulted in a lack of horizontal coordination [21–24]. This trend has negatively affected the public sector, increasing the costs, reducing efficiency in the delivery of services and producing a loss of credibility in public institutions [14].

Although there are recent publications linking sustainability and organizational change [25–27], the field is still underresearched. In particular, there is insufficient knowledge regarding the way organizational change linked to sustainability could be initiated, implemented and institutionalized in practice [28]. Indeed, although an array of tools and frameworks has recently been developed to support organizations in engaging with the SDGs, i.e., [29–33], these essentially refer to the initial steps ("problem definition" and "goal setting") or the final ones ("mapping" and "reporting") of a strategic management process, and they do not support actual strategy development, the stage in which organizational change should be undertaken [8]. This paper addresses this research gap by adopting a case-study approach. Our study focuses on the initial stages of the 2030 Agenda implementation in a public organization. This paper aims at shedding some light on the way public sector organizations can start the implementation of the SDGs. We intend to answer the following research question: How can the process of SDGs prioritization be used to promote internal collaboration and prepare the organization to face the organizational changes needed to adequately address the SDGs?

Our point is that, if seen through the lens of organizational change theory, the prioritization process could be useful in identifying necessary structural changes to address the SDGs and in establishing a diagnosis regarding organizational readiness to embrace change [34], assessing existing restraining and driving forces for undertaking this change [35] and promoting horizontal collaboration within the organization. Moreover, this paper describes a case study of particular relevance, which could be useful for other public sector organizations starting the implementation of the SDGs.

The case study presented here, based on the experience of the former Spanish Ministry of Agriculture, Fisheries, Food and Environment (MAPAMA), provides analytical material of high quality and interest since MAPAMA was a public organization at the highest level within the public administration structure and was responsible for many transversal aspects related to sustainability. MAPAMA, now divided into two ministries—the Ministry of Agriculture, Fisheries and Food (MAPA) and the Ministry for Ecological Transition and Demographic Challenge (MITECO)—had key competences over many of the areas represented in the SDGs, such as climate change, terrestrial and marine biodiversity and food systems, among others. Furthermore, it was one of the first public-sector

organizations in Spain that underwent a participatory process involving Ministry officials in order to interpret and prioritize the SDGs. The prioritization exercise reported in this paper was indeed a pioneer experience, whose results later contributed to the action plan for the implementation of the 2030 Agenda in Spain [36].

The article is organized as follows: Section 2 develops a theoretical overview focusing on the organizational challenges presented by SDG implementation in the public sector and the change management frameworks used in the case study. Section 3 presents the research methodology and the case study context. Section 4 presents the main results of the case study from an organizational perspective. Section 5 discusses the findings, highlighting the aspects that could be more easily applicable to other public sector organizations. Finally, Section 6 presents some conclusions and opportunities for further research.

## 2. Theoretical Overview

Along with intersectoral coordination, the call to transformation of the Agenda imposes the need for an increased coordination within the public sector itself, as the SDGs require the implementation of complex and transversal policy instruments that cut across different public organizations, administrative levels and policy areas [37]. However, achieving appropriate levels of inter- and intrasectoral coordination creates considerable difficulties. The public sector literature has consistently suggested that the lack of coordination and coherence across governmental departments or programs is one of the major challenges for contemporary governments [4,14,38–40]. Authors have used terms such as "vertical silos", "tunnel vision" or "departmentalism" to refer to a lack of horizontal coordination in this context [22–24]. Problems of coordination in the public sector translate into increased costs, reduced efficiency in the delivery of services and a loss of credibility of public institutions [14]. These problems are, to an important extent, linked to the wave of reforms implemented in many countries around the new public management phenomenon, which has resulted in an increased specialization, decentralization and fragmentation of the public sector, as well as in the proliferation of autonomous agencies [21]. The need for a "more horizontal" government is also connected to a number of contemporary forces or trends, such as the increasingly international dimension of public policy, or the need to achieve a better "customer orientation" by creating programs around specific "client" groups (women, families, the elderly, etc.), rather than developing programs focusing on the expertise or competences of a specific department [14]. Finally and most importantly for this work, a more coherent and horizontally-coordinated government is considered essential in order to handle the so-called "wicked" problems; i.e., problems that, as all those addressed by the 17 SDGs, are hard to define, are affected by high levels of uncertainty, have multiple solutions and cannot be approached from a single public organizations or policy area [41–43].

The SDGs represent a specific challenge for coordination. The SDGs were agreed on by all United Nations Member States in September 2015, when the 2030 Agenda was adopted. The Agenda comprises 17 goals (SDGs) and 169 targets (SDG targets) that integrate the three dimensions of sustainable development—economic, social and environmental—and seek to promote a transformation towards sustainability, following the "leaving no one behind" principle [44]. The SDGs are interdependent, as human, technical and natural systems are deeply interconnected [45]. The SDGs not only require collective action at scale among different sectors, organizations and disciplines but also social participation [46]. Although the SDGs represent an aspiration, they do not provide the pathway to get there; innovation and creativity are needed to address problems that do not have a predefined solution [46]. The public sector needs to coordinate vertically and horizontally in a multilevel governance, and to be open to participation of stakeholders, to ensure policy coherence and provide integral responses to the SDGs [45,47]. Specifically, public organizations play an important role translating global aspirations embedded in the SDGs to national contexts, acting as a model of adoption of the SDGs into their strategies and focusing on the common good [11].

Public administration literature distinguishes three main approaches to coordination: hierarchical, market and network [48]. In the hierarchical approach, coordination is imposed through traditional authoritative governance mechanisms. The market approach considers how coordination results from competitive dynamics consisting of exchange and bargaining among service "buyers" and "sellers" with somewhat opposing interests. Finally, in the network approach, coordination is achieved through collaboration dynamics among several legally autonomous entities that work together to achieve not only their own goals but also a collective goal [49]. Several authors claim that, in order to appropriately address the SDGs, a combination of hierarchical, market and/or network approaches is needed [50,51]. Within this framework of governance, or "metagovernance", these apparently contrasting approaches are mutually enforcing and contingent to the particular policy conditions [15].

Regardless of the strategy considered, the implementation of coordination mechanisms in the public sector is a complex endeavor, which often requires important structural and behavioral changes at the design and implementation levels [14,15]. In particular, the implementation of horizontal collaboration and network strategies requires time and a development of collaboration skills [15]. It also requires an appropriate support and mindset at the leadership level [41], i.e., new models of leadership that better appreciate the distributed nature of information, interests and power. Senior officials need to go beyond the traditional notion of a top-down direction and rather lead organizational members into collective reflection processes [52]. It should be noted that agents involved in horizontal collaboration need to be allowed an appropriate level of autonomy to engage in reciprocal exchanges and build trustful relationships. Institutional contexts and incentive schemes also need to be adapted to accommodate collective objectives and motivate collaborative behavior [53].

From the literature of change management, it is known that different sorts of blocking mechanisms and resistance to change are likely to arise before all these organizational changes are consolidated [54,55]. To avoid these blocking mechanisms, it is necessary to anticipate and identify potential resistances and catalysts to change in the initial steps of implementation [56,57]. Typically, the process of organizational change comprises three phases. Lewin (1951) [35] considers the change process to involve unfreezing, moving and refreezing. Other authors have used other terms, such as readiness, adoption and institutionalization of change [58]. When the forces against change are equal or stronger than the forces driving the change, the organization is frozen in an equilibrium state. For an organizational change to happen, the organization must enter an unstable stage ("unfreeze" stage) where forces for change are activated and capable of overcoming forces against change. Armenakis and Harris (2002) [34] refer to this phase as the "readiness" stage, where organizational members become prepared for the change and ideally become its supporters. The concept of readiness for change has been extensively used in change management literature [59–62]; it can be defined as "a collection of thoughts and intentions towards the specific change effort" [63] (p. 40). In the second phase, the change is implemented, and different ways of operating are adopted. Finally, in the third phase, the organization arrives at a new equilibrium state ("refreeze" or "institutionalization" stage) where the desired change is adopted and becomes internalized.

The diagnosis of catalysts and barriers for change can be conceptualized using the force-field approach [35]. Lewin's model understands behavior in an institutional setting not as a static (motionless) habit or pattern but as a dynamic balance of forces working in opposite directions. According to this way of looking at patterned behavior, change takes place when an imbalance occurs between the sum of the forces against change (restraining forces) and the sum of the forces for change (driving forces). A force-field analysis assumes that any social situation is a balance between these forces. An imbalance may occur through a change of magnitude or a change in direction in any one of the forces, or through the addition of a new force. A force-field analysis is useful for diagnosing a problem and helps to identify forces operating on both sides of an issue. This model has been extensively used in different contexts and disciplines, and in particular in studies related to collaboration (i.e., [64]).

We suggest using the prioritization process, as we consider it a good occasion to identify new coordination needs associated to SDG implementation and to establish a diagnosis regarding possible

levers and barriers to change. The prioritization process of the SDGs intends to identify a reduced set of priorities [7] so that the organization can introduce them into its strategy and therefore facilitate organization decision-making towards the implementation of the priorities [65]. The development of a strategy, together with the decision to allocate resources for its execution, form what is called strategic management [66]. Grainger-Brown et Malekpour (2019) [8] highlighted that most strategic management processes share three phases: the ideation phase, the development phase and the implementation phase. Although a technical exercise of prioritization is more related to the ideation phase—where strategic goals are defined and articulated—the introduction of an organizational perspective that identifies coordination mechanisms and foresees internal forces for and against change is more related to the development phase, where the different pathways to meet the prioritized SDGs are visualized and evaluated. Therefore, the introduction of an organizational perspective to the prioritization process may contribute to connecting the ideation phase of the strategic management with the development phase and to promoting strategic actions that link organizational change to sustainability, a field that is still underresearched [28].

## 3. Research Approach

### 3.1. Research Methodology

This investigation is approached through a case study methodology. A case study is typically used to investigate a contemporary phenomenon (i.e., "the case") in depth and within its real-world context. As they are based on a variety of data sources, case studies offer rich empirical descriptions of particular instances of a phenomenon [67]. The methodology is particularly applicable when the research questions can be formulated in terms of "how" and "why". Over the last decades, case studies have been used extensively in multiple fields, including organizational theory [68], strategy and decision science [69] and, most importantly for this work, sustainability [70–72]. In particular, case studies have been used in previous works that explore the prioritization process of sustainability aspects within organizations [10,11,73]. Case studies can be used for descriptive purposes, as it is the case in our paper, but they can also play an important role in theory testing and building [74]. It is worth noting that part of the research team that developed the case was directly involved in the actual prioritization process in various capacities and worked collaboratively with key personnel at MAPAMA to develop a diagnosis. The research team also included external researchers. Hence, this study's methodology can also be considered as collaborative management research, i.e., an effort by two or more parties, where at least one of whom is a member of an organization or system under study and at least one of whom is an external researcher, to work together in learning and producing the necessary information [75,76]. In this type of research, the researcher is not a mere observer but an agent of change who engages in cogenerating "actionable scientific knowledge" [77,78].

The case study was conducted between September 2016 and December 2017. However, the analysis regarding the need for internal collaboration and organizational readiness for change was added to the study at a later stage, between March and July 2020. Figure 1 presents the overall research process. The methodology applied comprised three phases: prioritization, participative diagnosis and validation.

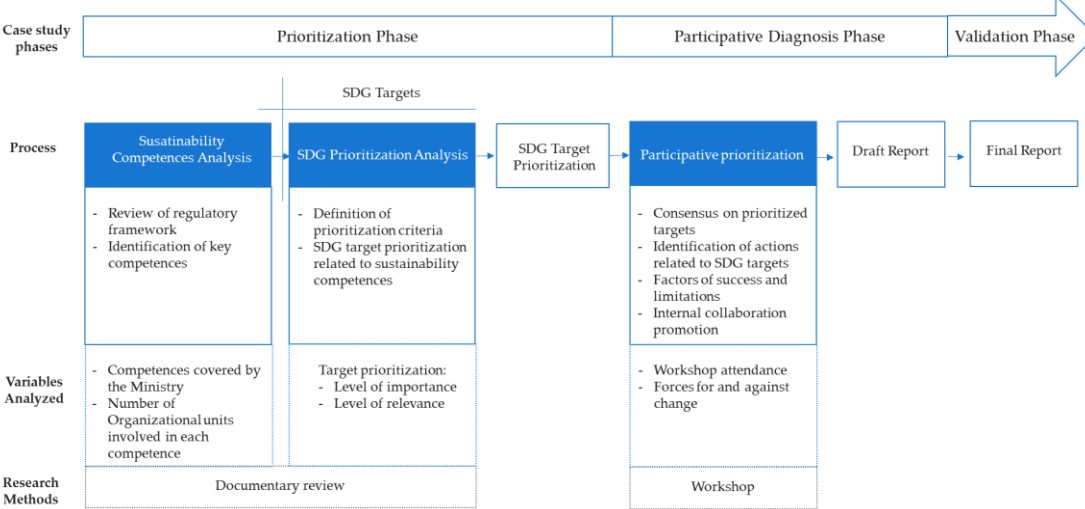

**Figure 1.** Overall research process.

Prioritization took place between September and October 2016, where four work meetings were held between the research team and selected ministry officials. First, based on the regulations in force and the Ministry's organizational structure, an analysis for identifying the sustainability competences attributed to each organizational unit was made (i.e., the sustainability competences analysis). Second, an analysis of the potential contribution of the Ministry upon each SDG target according to their level of importance and responsibility was made (i.e., the SDG prioritization analysis). Together, the sustainability competences analysis and the SDG prioritization analysis showed the potential contribution of each organizational unit to each SDG target. These analyses were developed by the research team through a documentary review method. The following sources of information were used during the documentary review: key documents associated with the competences of the Ministry (including plans and programs, strategies, policies, legislation, European Union regulations, etc.); key documents associated with SDG implementation available at the time of the first stage of the case study; and primary information produced by the researchers.

Participative diagnosis took place between October 2016 and June 2017. Based on the potential contributions of each organizational unit to the SDGs identified in the previous step, nine thematic workshops were held. Each thematic workshop aimed at discussing prioritized SDG targets, identifying current actions linked to each SDG target as well as factors of success and limitations, collecting suggestions, and ultimately, promoting internal collaboration among organizational units involved in the process. The participative diagnosis involved 89 Ministry officials from 16 Directorates-General through a workshop method. A workshop is an arrangement whereby a group of people learn, acquire new knowledge, perform creative problem-solving or innovate in relation to a domain-specific issue [79]. It is designed and conducted by people with experience with the domain to fulfil a predefined but not predictable purpose; participants are thus expected to actively participate and influence the workshop direction. Workshops are widely acknowledged as a participatory research methodology [80] in different disciplines, including organizational change [81]. In this research, the workshops were organized by the research team, and each workshop included at least two facilitators and one additional researcher reporting contributions. All workshops followed the same structure, and the information gathered was analyzed using coded-based content analysis techniques [82,83]. The results of the diagnosis were collected in a draft report where each SDG target was classified according to its level of priority, global value, current state, evolution, alignment and coordination needs. Appendix A compiles further information about the context and structure of the workshops.

Validation took place between June 2017 and December 2017, and it included the presentation of the draft report to all organizational units in the workshops, the collection of comments from the organizational units on the SDG targets diagnosis and the conclusions and recommendations.

*3.2. Context of the Case Study*

In 2016, the implementation phase of the 2030 Agenda began, and private and public organizations started considering the goals and targets where they could have greater contributions based on their business models and competences. Inside the Spanish public administration, the former MAPAMA was one of the first organizations at the national level undertaking a process to implement the SDGs. In September 2016, MAPAMA started a prioritization process in order to identify the SDGs that were most relevant to the Ministry as the basis for the development of a strategy for institutional alignment with the 2030 Agenda. The facilitation of this initiative was commissioned by the Ministry's subdepartment of the International Relations and Community Affairs (RIAC) to the Innovation and Technology for Development Centre of the Universidad Politécnica de Madrid (itdUPM). As presented in Appendix A, MAPAMA is organized according to 16 Directorates-General that belong to 4 different Secretariats. For the purposes of our study, from now on Directorates-General are referred to as "organizational units", while Secretariats are referred to as "superior organizational units".

## 4. Results

The work was considered a successful process due to, among other aspects, the participatory approach introduced, involving a considerable number of Ministry officials in the diagnosis of MAPAMA's contributions to the 2030 Agenda. The success of the work was mentioned by a high official of the Ministry in the Spanish Senate [84], and it served as a model to other Spanish public administration to configure the action plan for the implementation of the 2030 Agenda in Spain that was presented at the High-Level Political Forum of the United Nations in 2018 [36]. The facilitating role played by the RIAC was crucial to this success, as it ensured communication channels with all areas of the ministry.

The following subsections provide a more detailed explanation of the main results of the process: the sustainability competences analysis, the SDG prioritization analysis and the organizational units' participation during the participative diagnosis.

*4.1. Sustainability Competences Analysis*

The Spanish public sector is highly regulated, and national and international regulations define the set of responsibilities of each administrative body, known as competences. Furthermore, the Spanish administrative organization creates a need for coordinated action between different levels of public administration: the general state, the autonomous communities and the local level [85]. In this line, the Spanish Constitution establishes the competences that are exclusive to the State and the competences that could be delegated to the autonomous communities [86]. This encourages cooperation and collaboration between national and regional governments [87], but also makes it more difficult to clearly define the competences that correspond to a given ministry [88]. In particular, MAPAMA is subject to an extensive regulation regarding its field of competence and current responsibilities. Identification of competences related to sustainability was mainly based on the 401/2012 Royal Decree [89], which established the competences of the ministry. Furthermore, to ensure the connection between sustainability competencies and the areas of the ministry responsible for them, the organizational structure was analyzed. The regulation applying to MAPAMA's structure was the 424/2016 Royal Decree [90]. This Royal Decree established the organization of the Ministry into different Directorates-General with specific responsibilities associated to them. This analysis led to the identification of sustainability competences connected to specific organizational units. Table 1 shows a synthesis of the analysis made, showing the sustainability competences identified and the number of organizational units responsible for each competence.

**Table 1.** Sustainability competences and number of responsible organizational units.

| Sustainability Competences with One Responsible Organizational Unit | Sustainability Competences with Two Responsible Organizational Units |
| :---: | :---: |
| Sustainability training | |
| International agreements and negotiations | |
| Multilateral organizations | |
| Organic farming | |
| Agricultural trade | |
| Food industry | |
| Food quality and safety | |
| Agricultural diversity and patents | |
| Irrigation | |
| Forestry | |
| Land ownership and farm structure | |
| Sustainable rural development | Strategic planning and management |
| Territorial planning | Food chains |
| Environmental education | Animal and plant health |
| Environmental audit and responsibility | Fertilizers and phytosanitation |
| Chemical products | Food production and agricultural markets |
| Waste and contaminated soils | Fishing and aquaculture |
| Industrial environment | Coastal and marine environment management |
| Environmental impact assessment | Protected spaces management |
| Air quality and noise | Biodiversity conservation |
| Sustainable water management | Climate change awareness |
| Climate change adaptation | |
| Climate change mitigation | |

*4.2. SDG Prioritization Analysis*

Once the sustainability competences for each organizational unit were identified, a prioritization analysis regarding their potential contribution of the Ministry upon each SDG target was made. Potential contribution was assessed according to two variables: relevance of the SDG target according to public policies currently in force and the level of responsibility of the Ministry towards each SDG target. SDG targets were codified according to a three-level potential contribution scale: strong, medium and weak. Strong potential contribution was assigned to those competences in which the ministry had a responsibility for direct action or execution on the SDG target. Medium potential contribution was assigned to competences in which the ministry had a responsibility limited to action in specific areas that contributed to the SDG target, such as monitoring and awareness raising. Finally, weak potential contribution was assigned to competences in which the Ministry takes into account the SDG target but does not impact on it directly. Figure 2 shows the summary of MAPAMA's potential contributions to SDG targets. Out of the 137 SDG targets considered as potential contributions, 62 of them were categorized as strong contribution, 26 as medium contribution and 49 as low contribution. In terms of distribution, strong contribution appeared in 100% SDG targets of SDGs 2, 6 and 15 and 40%–90% SDG targets of SDGs 7, 12, 13 and 14. In contrast, 10% or less SDG targets of SDGs 3, 4, 5, 10, 16 and 17 were strong contributions.

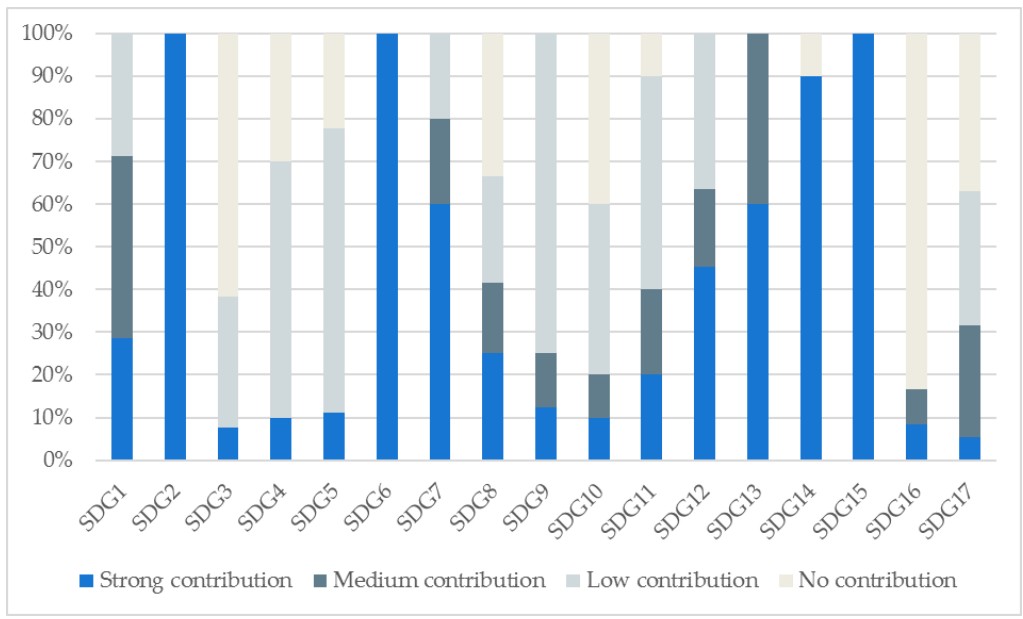

**Figure 2.** Summary of the Sustainable Development Goals (SDG) prioritization analysis.

### 4.3. Organizational Unit Participation in the Workshops

Nine thematic workshops were organized for an in-depth analysis of the SDG targets that were categorized as "strong potential contributions" in the SDG prioritization analysis and were for promoting transversal participation. Workshop topics were chosen considering the interrelated nature of the SDGs, and according to the sustainability competences analysis, the organizational units related to selected SDG targets were invited to participate in the corresponding workshops. Although all organizational units involved in the contribution to the SDG targets addressed in each specific workshop were invited, not all of them could attend. Figure 3 compares the number of organizational units invited with the number of units that attended and participated in the workshops. Overall, 63% of the organizational units participated in the workshops, and all organizational units participated in at least one workshop. It can be noted that those workshops in which the number of units convened was lower (Workshops 3 and 4) had a better participation rate than those that convened with greater diversity. In addition, all the workshops had representation from at least three different units, and a total of 112 Ministry officials participated in the thematic workshops (89 different Ministry officials, of whom 61 were women and 28 men). Moreover, 12 Ministry officials participated on average in each workshop, with Workshop 9 having the most participants from the Ministry (18 Ministry officials attended) and Workshops 4 and 7 the least (8 Ministry officials attended). The high number of participants and units involved in meetings was considered by the Ministry officials as one of the intrinsic successes of the study.

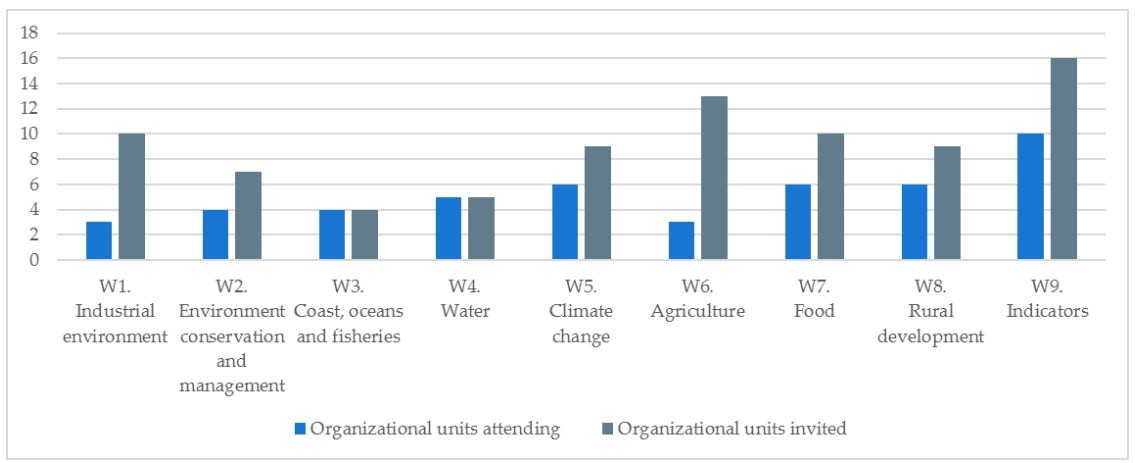

**Figure 3.** Participation of organizational units in each workshop.

### 4.4. Forces of Change Identified during the Workshops

The thematic workshops also allowed for the detection of perceived barriers and drivers for internal collaboration. The participants' perceptions were conceptualized using an "organizational readiness" lens [34]. As explained in the theoretical framework, organizational readiness refers to an initial stage within a change process, where Ministry officials in the organization become prepared for the change. In this case, the needed organizational change is a better and more fluid internal collaboration around the prioritized SDG targets, many of them highly transversal in nature. The minutes of the workshops were codified using a force-field analysis approach. Figure 4 summarizes the main drivers and barriers for collaboration detected as well as the number of times they were mentioned by the participants in the workshops (recurrence).

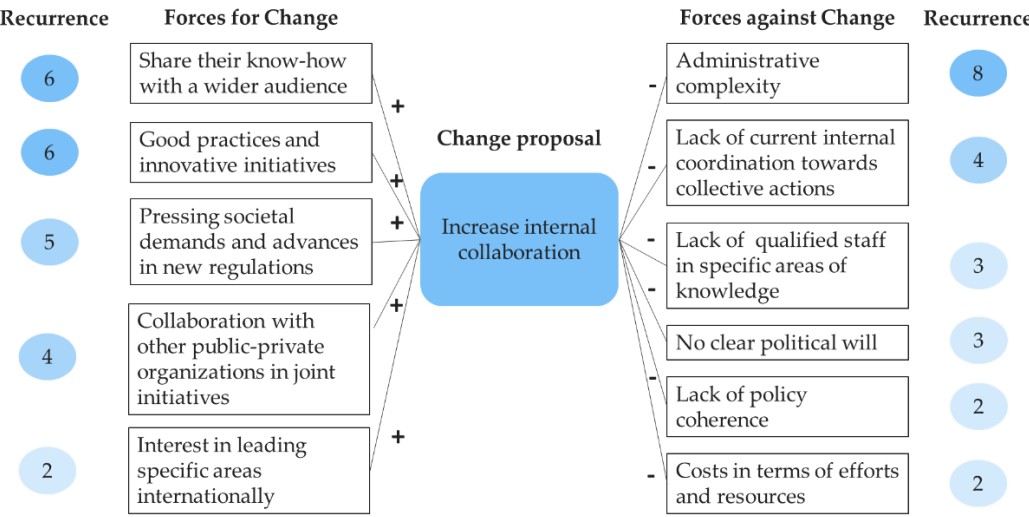

**Figure 4.** Summary of restraining and supporting forces.

A number of context factors were identified as enablers for change. The most common driving force was the feeling of motivation related to being involved in a pioneering process in the Spanish public administration related to sustainability and the SDGs, as well as the excitement of sharing their know-how and performance with a wider audience. Another motivation factor was the possibility of identifying good practices and the associated improvement opportunities. The group acknowledged a good disposition for sharing information with the facilitators of the workshops, which suggests that an external organization could play an enabling role to progress on the proposed change [59]. Reinforcing internal collaboration was seen as the right step to make, taking into account the demands

of different stakeholder groups in topics such as the improvement of recycling practices or climate change, as well as recent regulations that call for collaboration of different organizational units as well as other ministries, such as a new regulation on circular economy or the design process of climate change policies. In addition, the development of collaboration with other public and private organizations in joint initiatives, i.e., the initiative "ReEducamar" [91], which is the network and inventory of marine education resources in Spain aiming at promoting awareness regarding sea conservation, was highlighted as a good practice and a possible benchmark promoting internal collaboration. In the same vein, the opening of spaces where the Ministry interacts with specific groups of society interested in areas of the SDGs related to MAPAMA competences was seen as a good practice that reinforce internal collaboration to respond to the demands of society. Finally, the interests of MAPAMA for achieving international leadership in certain areas through a pioneering effort in implementing the SDG were also identified as drivers for change.

Regarding the barriers, the most common restraining forces mentioned to increase internal collaboration were the complexity and fragmentation of the Spanish public sector and in particular the division of responsibilities between the regional and the national levels. These are classic issues that public organizations face when trying to work more collaboratively [4,14,38–40]. Moreover, many participants declared that the workshops had enabled them to identify "potential synergies between the missions carried out in different organizational units of the Ministry, and even between MAPAMA and other public administrations at the national or regional level". A need for deepening scientific and technical knowledge in key areas of the Agenda was identified by participants as an important aspect to address the SDGs. No specific needs regarding competences or capacities for enhancing internal coordination were explicitly mentioned [15], although participants pointed out the lack of time to develop such collaboration. Despite the fact that the diagnosis process was supported by a high-level official in the Ministry and the head of the Ministry, who showed their commitment to the 2030 Agenda in public appearances, participants did not acknowledge a clear political will to promote more internal collaboration nor particular incentives towards that goal. In addition, related to ideological differences among administrations, participants highlighted the lack of coherence in policies that were developed by parties with different approaches. Finally, participants perceived that increasing internal collaboration would imply high costs in terms of effort and resources. As mentioned in the theoretical overview, to compensate for these costs, literature suggests boosting participants' ownership to foster enduring and effective coordination [14,15].

## 5. Discussion

### 5.1. Need for Internal Collaboration and Coordination Approaches

The sustainability competences analysis allowed the different organizational units of MAPAMA to have a clear picture as to what their part of responsibility is. Furthermore, the combination of the sustainability competences analysis and the prioritization enabled assessing to what extent the prioritized SDG targets were transversal across the organization, allowing the organizational units to be aware of their relationship with specific SDG targets and having a clear map of where each one contributes to in the 2030 Agenda and who is contributing with them. Figure 5 presents the degree of transversality of SDG targets categorized as "strong contribution", measured through the number of organizational units involved in the achievement of specific SDG targets in various capacities. The SDG target transversality provides information about the need for collaboration among organizational units to address specific SDG targets. In the case of MAPAMA, out of the 62 SDG targets in the "strong contribution" category, 15 SDG targets were transversal to 2 organizational units, and 29 SDG targets were transversal to 3 or more units, with SDG target 12.4 being the most transversal (demanding the collaboration of 10 organizational units). SDG transversality is useful to show Ministry officials what their contribution to the "bigger picture" is from their position and who they should collaborate

with to address the SDG target in an integrated way, which is one of the recommendations for the success of public sector SDG implementation [1].

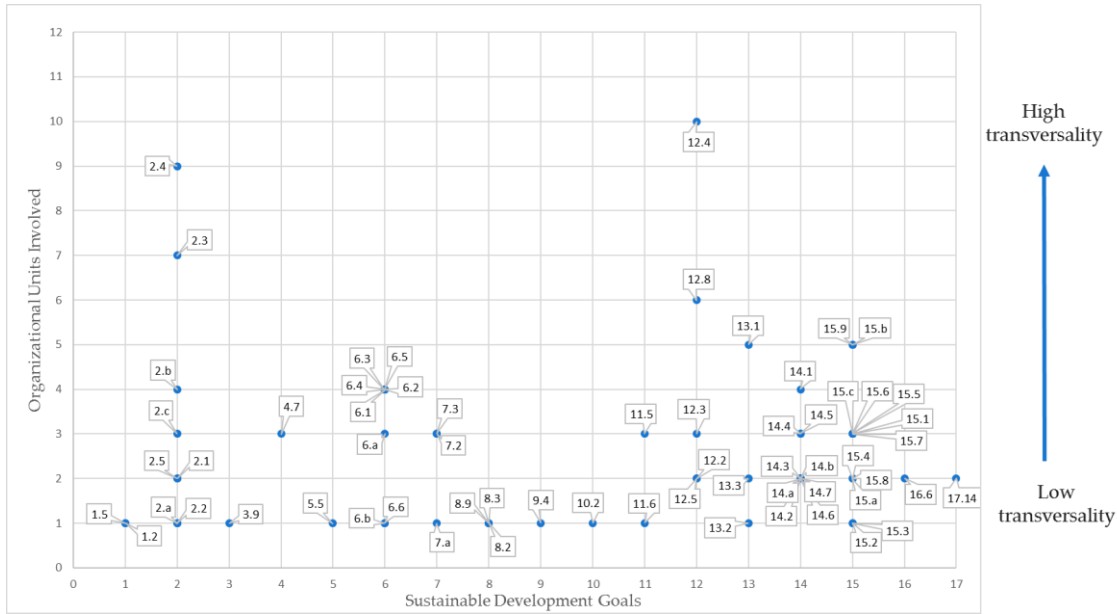

**Figure 5.** Transversality of SDG targets categorized as "strong potential contribution" for MAPAMA (the Spanish Ministry of Agriculture, Fisheries, Food and Environment).

Regarding the coordination approaches, the SDG targets that are not transversal for organizational units will not require changes in the current hierarchical structure. In contrast, more transversal SDG targets will likely represent a bigger challenge in terms of establishing new coordination strategies [22–24]. To face this challenge, changes in the current coordination approach should probably be introduced to develop new network coordination mechanisms that promote internal collaboration and overcome the "silo" logic.

Presumably, the more transversal a SDG target is for the organization, the bigger the coordination effort is necessary. For instance, achieving internal coordination to address SDG target 12.4, which involves 10 organizational units, would be more difficult than achieving internal coordination for SDG target 7.2, which involves 3 organizational units only. Along with the number of units involved, it is also necessary to consider preexisting horizontal collaboration mechanisms. In the case of MAPAMA, SDG target 12.3 is supposed to involve less effort in terms of development collaboration strategies than other SDG targets involving the same number of units, as involved units have already been collaborating in the framework of a working group around the policy of food waste since 2014. Finally, since the network approach will coexist with the hierarchical approach, the position inside the hierarchy of the organizational units will also influence the effort of introducing the new approach. As an example, SDG target 12.3, which is transversal to three organizational units that belong to the same superior organizational unit according to MAPAMA organigram, will suppose a lower challenge than SDG target 15.c, which is also transversal to three organizational units but is dependent on three different superior organizational units within the Ministry.

In conclusion, organizations must combine different coordination approaches, according to the demands that each specific SDG target makes upon the organization. If an SDG target cannot be addressed integrally through the current hierarchical structure, a process of organizational change is needed in order to face the identified organizational silos and promote a new approach of horizontal coordination. The introduction of new collaboration mechanisms will require heterogeneous efforts depending on the level of transversality of the SDG targets, the previous internal collaboration experiences and the coherence between the hierarchical structure and the new collaboration needs.

*5.2. Readiness for Change*

In the previous section, an emphasis on the need for establishing appropriate coordination mechanisms around transversal SDG targets was made. However, structural changes are not enough, and changes in behavior are essential to make the coordination successful [14,15]. The information gathered during the thematic workshops developed as part of the field work allowed for a better understanding of the position of the Ministry officials involved and their attitude towards change. Although the explicit goal of the thematic workshops was to discuss and validate the prioritization proposed, the participatory dynamics of the workshops were also an opportunity for the research team to capture the units' disposition towards collaborating around transversal SDG targets and the participants' perception of internal barriers and levers to collaboration.

Analyzing the participation rate of the organizational units during the thematic workshops may give some clues about where the strategy development can start more efficiently and successfully. For example, the high participation rate of Workshops 3 and 4—with 100% participation rate of units involved—can be interpreted as a disposition of the organizational units involved to work collaboratively towards the achievement of SDGs related to coasts, oceans and fisheries (Workshop 3) and water (Workshop 4). Therefore, these areas may be potential first runners in the introduction of horizontal coordination mechanisms and secure a short-term win in the proposed change, which is important in the first stages of organizational change [56]. On the contrary, addressing horizontal collaboration in SDGs related to environmental industry and agriculture (Workshops 1 and 6, respectively) may be more time-consuming and may require more resources to produce changes in the expected direction.

In addition, focusing on the results of the forces for and against change, participants can detect elements that can act as enablers or barriers in the different pathways that the organization could take in the strategy development phase. For example, the lack of scientific and technical knowledge in key areas of the Agenda was perceived as a barrier to adequately address the SDGs, while collaboration with other public and private organizations in joint initiatives was highlighted as a good practice. Accordingly, a pathway to achieve the needed deep knowledge in key areas could come from a strategy of fostering partnerships with other organizations.

It is worth noting that all organizational units in the Ministry participated in the process to some extent. Literature has highlighted that this is important to create a collective readiness of change that promotes group sensemaking and a shared belief towards change, which are important elements in developing a multilevel perspective of change [59]. Kim et al. (2011) [92] considered participant engagement in these types of participatory processes an example of change supportive behavior, i.e., "actions employees engage in to actively participate in, facilitate, and contribute to a planned change initiated by the organization" [92]. For this to happen, the role of senior officials was relevant, fostering communication between organizational units and encouraging attendance at workshops. As an example of this, it is worth mentioning that these workshops were activated due to one of the senior officials of the MAPAMA from the Secretariat of Agriculture and Fisheries, Food and Environment. This official, knowing the direction that the organization should take (i.e., the adoption of the SDGs), commissioned this case study to the Technology for Development Centre of the Universidad Politécnica de Madrid (itdUPM) through his subdepartment of the International Relations and Community Affairs (RIAC); the itdUPM was in charge of promoting a bottom-up process supporting distributed leadership and leveraging the collective intelligence of the organization.

To sum up, the introduction of the organizational perspective contributed to connecting the prioritization process with the development of the strategy. Firstly, it allowed for identifying the need for new coordination mechanisms to work adequately around highly transversal goals. Secondly, it provided a diagnosis of catalysts and barriers for organizational change. This information is useful to decision-makers of the organization in choosing specific pathways to implement a strategy around SDG implementation. Finally, validating the priorities with internal agents and setting up a participative

process helps with envisioning the collaboration dynamics that the different organizational units should reproduce when implementing the strategy.

## 6. Limitations, Further Research and Conclusions

### 6.1. Limitations and Further Research

In-depth single case studies are appropriate when studying new or emergent fields, such as SDG implementation in the public sector, as they allow the researcher to have a deeper understanding of the explored subject [93–95]. The main conclusion of this study, in particular the idea of using the prioritization phase as a lever for initiating strategy development in a context of SDG implementation, is highly applicable to any public sector organization. None withstanding, further studies in different national and regulatory contexts would be useful and would enable contrasting the applicability of results.

Transversality analysis has been useful in detecting organizational change needs at the level of a single public organization; conducting this same analysis at the level of the whole public administration would allow for a more integral perspective, putting into practice the "whole of government" approach to link different goals together and overcome trade-offs [1,14,83,84]. For example, prioritized SDG target 5.5 on gender is not very transversal in MAPAMA, but if other administrations' perspectives were introduced in the analysis, this SDG target might be one of the most cross-sectional targets within Ministries, as it is identified as an SDG target that will require high levels of interministerial collaboration. In addition, in this research, transversality analysis was limited to those SDG targets ranked as "strong contribution". Further research should apply transversality analysis to all prioritized SDG targets.

This study focused on establishing a diagnosis regarding both the organizational changes needed and the organizational readiness. Further research should concentrate on establishing appropriate strategies for creating organizational readiness and unlocking change [63]. As an example, a key element in the readiness for change perspective is the articulation and communication of an appropriate change message [34,63]. Further research in this direction is suggested, namely in interpreting force-field analysis findings in relation to the five dimensions for crafting a change message: (a) discrepancy, which is the sentiment regarding whether the change is really needed; (b) efficacy, the sentiment regarding one's ability to succeed; (c) appropriateness, a sentiment about the intended change being appropriate in a specific institutional context; (d) principal support, a sentiment about having enough resources and support for implementing the change; and (e) personal valence, the perceived benefits of disadvantages of the change for the individual [34,63].

Finally, in June 2018, the newly elected government of Spain decided to separate MAPAMA into MITECO and MAPA. In future research works, it would be interesting to delve into the implications of this structural change in terms of the coordination between the new two ministries to contribute to the SDGs.

### 6.2. Concluding Remarks

For public organizations, applying structured approaches for prioritization is important in the initial steps of strategy development as they enable concentrating on a set of priorities, but this is not enough to ensure that these priorities are effectively implemented.

The introduction of an organizational perspective during the prioritization process is key to supporting SDG implementation and to boosting the transformative capacity that underpins the 2030 Agenda. The approach followed in this case study could be useful for other public actors and agencies elsewhere that are in the early stages of strategy development regarding the SDGs, particularly complex administrations with a highly hierarchical structure. In the context of this case study, it helped to identify collaboration needs in addressing the SDGs, and this enabled the identification of driving and restraining forces in initiating the required organizational changes. Furthermore, setting participative

processes that engage internal agents around the development of the implementation is essential not only to identifying the elements that are key to unlocking change potential in future stages, but also to reproducing the dynamics of internal collaboration that will be needed for the adoption of the 2030 Agenda.

**Author Contributions:** Conceptualization, M.S., T.S.-C.; methodology, M.S., T.S.-C., J.U.; validation, J.U., D.P.; formal analysis, M.S., T.S.-C.; investigation, M.S., T.S.-C.; writing—original draft preparation, M.S., T.S.-C.; writing—review and editing, M.S, T.S.-C., J.U., D.P.; visualization, M.S.; supervision, T.S.-C., J.U., D.P.; project administration, M.S., J.U., D.P.; funding acquisition, M.S. All authors have read and agreed to the published version of the manuscript.

**Funding:** The project corresponding to this case study was funded by MAPAMA. This doctoral research was supported by funding from the Universidad Politécnica de Madrid (UPM).

**Acknowledgments:** The authors would like to acknowledge the hard work of MAPAMA personnel supporting the prioritization, diagnosis and validation processes, especially those from the Subsecretary of Agriculture, Fisheries, Food and Environmental of MAPAMA: Marta Cimas, Gonzalo Eiriz, Marta Dopazo, Jesús González, Marina Álvarez and Maria del Carmen González. Moreover, we would like to acknowledge the efforts of the other members of the project team from the Universidad Politécnica de Madrid: Carlos Mataix, Inma Borrella, Marina Moreno and Carlota Tovar. The authors would also like to acknowledge the indispensable contribution of all Ministry officials involved in the process.

**Conflicts of Interest:** The authors declare no conflict of interest. The funders had no role in the design of the study; in the collection, analyses, or interpretation of data; in the writing of the manuscript, or in the decision to publish the results.

## Appendix A

**Table A1.** Workshop Context.

| Meeting | Date | Thematic | SDG Addressed | Participants |
|---|---|---|---|---|
| 1 | 11 November 2016 | Industrial environment | SDG 9, SDG 11 | 13 |
| 2 | 15 November 2016 | Environment conservation and management | SDG 15 | 10 |
| 3 | 17 November 2016 | Coast, oceans and fisheries | SDG 14 | 17 |
| 4 | 23 November 2016 | Water | SDG 6 | 8 |
| 5 | 24 November 2016 | Climate change | SDG 13, SDG 7 | 15 |
| 6 | 29 November 2016 | Agriculture | SDG 2, SDG 12 | 12 |
| 7 | 1 December 2016 | Food | SDG 2 | 8 |
| 8 | 3 December 2016 | Rural development | SDG 5, SDG 8, SDG 10, SDG 17 | 11 |
| 9 | 7 February 2017 | Indicators | SDG indicators | 18 |

**Table A2.** Workshop structure.

| Time | Activity | Participants' Role |
|---|---|---|
| 30 min | Introduction to the workshop and the SDGs to be addressed | Listening to the presentation |
| 15 min | Analysis of actions related to specific SDGs | Individual work on filling out a template |
| 30 min | Collection of actions and validation of SDG prioritization | Presenting individual work and discussing in groups |
| 30 min | Analysis of drivers and barriers to achieve the specific SDGs | Group work on filling out post-its and commenting their perspective |
| 20 min | Sharing of group outputs | Presenting group outputs to all participants |
| 5 min | Conclusions | Listening to main workshop conclusions |

**Table A3.** Organization chart of the Ministry of Agriculture, Fisheries, Food and Environment.

| Ministry | Secretariats (Superior Organizational Units) | Directorates-General (Organizational Units) |
|---|---|---|
| Ministry of Agriculture, Fisheries, Food and Environment | | Cabinet Office<br>National Parks |
| | Secretariat of State for the Environment | Spanish Climate Change Office (OECC)<br>Quality and Environmental and Natural Environment Assessment<br>Coast and Sea Sustainability<br>Water<br>State Meteorological Agency (AEMET) |
| | Secretariat of Agriculture and Fisheries, Food and Environment | Services<br>Technical General Secretary |
| | General Secretariat of Agriculture and Food | Agricultural Productions and Markets<br>Health of Agricultural Production<br>Rural Development and Forest Policy<br>Food Industry<br>Agricultural Guarantee Fund (FEGA) |
| | General Secretariat for Fisheries | Fishery Resources and Aquaculture<br>Fisheries ordinance |

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
