# Peer review of "Introducing an Organizational Perspective in SDG Implementation in the Public Sector in Spain: The Case of the Former Ministry of Agriculture, Fisheries, Food and Environment"

_sustainability, doi:10.3390/su12239959_

Round 1
Reviewer 1 Report
This is a paper with high societal relevance and interest, very timely as well. Still, I would propose a couple of clarifications/elaborations in the paper to improve readability and understanding.
1) Theoretical overview: What is the role of public managers (senior officials) in this context. Later the story refers to "people" or "officials", but it is not clear how the internal hierarchies of the organisational units are considered. As it is linked to prioritisation, it cannot be a totally bottom-up story. I think it needs some comment (perhaps also in the empirical part of the case study (e.g. in 5.3), but also in the discussion readiness to change (6.2), where the position of public officials is mentioned, but in a different context (line 371)).
2) Case study:
If the general directorates are organisational units, then what means "superior units", this needs clarification. Maybe it can be done in Appendix A (if they can be identified there).
The numbers of priorities are slightly differently noted throughout the text and figures, for example, a) SGD12 and target 12, also b) 12.4 and 15.c. I would suggest harmonising a), and explain b). It would make things easier for the reader.
3) Discussion
The discussion around Figure 6 does not include better public service-related forces (expectations or demands of the general public or special interest groups of citizens representing the demand for these SGD related services and wider access etc.). This is something that has always been found to be one driver in public sector innovation literature. Can you comment on that in the paper, perhaps it is considered among the forces of change (e.g. good practices, advances of regulations)?
Reviewer 2 Report
Paper is very well written and interesting. Topic is relevant and worth in-depth studying. Authors delivered an organizational perspective on implementation of SDGs, relying on example of MAPAMA. Some minor changes, however, may improve the quality of the paper:
- introduction should more clearly state what is the objective of the paper and research; as for now, it seems to be an attempt to address research gap in respect of linking sustainability with organizational change;
- description of research methodology deserves more clarification too; authors stated that field study took place in years 2016-17, while the analysis "regarding the need for internal collaboration and organizational readiness for change were added to the study at a later stage" in fact, in 2020; it should be explained, what is the reason for such a three-year time gap and whether some assumptions has changed in the meantime; authors mentioned about participative group meetings as a way of collecting primary information - it should be distinguished between studying documents and interviewing/consulting experts (Delphi method maybe?); if there were workshops, were there any non-compliances of opinions, observations by participants? very little is known about the results of those workshops, in fact, except for generalized conclusions formulated by "participants";
- authors should briefly address genesis and key priority areas of SDGs;
- presentation of research results is mostly descriptive, more conceptual diagrams would be welcome;
- it would be interesting to mention whether division of MAPAMA into MAPA and MITECO impacted in any way the process of SDGs implementation, at least what was the reason (if any relevant) for such organizational change;
- concluding remarks should be extended to point at application value of research findings for the other public actors and agencies elsewhere, including drawbacks of case study.
Reviewer 3 Report
The presented article concerns the SDG implementation process in public sector entities in Spain, while the presented case study refers to the - no longer existing - Ministry of Agriculture, Fisheries, Food And Environment. The very fact that a unique unit (the only one in the studied area) was treated as a case study causes doubts as to whether the research results obtained in this unit can be generalized to other public sector units.
Regardless of the above, according to the Authors, the SDG implementation process can be treated as an element of change management in these organizations / this organization. In this context, they indicate the purpose of the study as “We intend to show that, if seen through the lens of organizational change theory, the prioritization process could be useful in identifying necessary structural changes to address the SDGs and establishing a diagnosis regarding organizational readiness to embrace change [34], assessing existing restraining and driving forces for undertaking this change [35], and promoting horizontal collaboration within the organization”. According to the authors, such a goal results from the identified research gaps, i.e.:
- “insufficient knowledge regarding the way organizational change linked to sustainability could be initiated, implemented and institutionalized in practice [28]”;
- and the fact that “tools and frameworks have recently been developed to support organizations in engaging with the SDGs, (…) do not support actual strategy development, the stage in which organizational change should be undertaken [8]”.
In the theoretical part of the article, the Authors show that the coordination deficits in public administration organization create considerable difficulties, which causes increased costs, reduced efficiency in the delivery of services, and a loss of credibility of public institutions. I do not quite understand why the Authors a fundamental problem related to the coordination of activities in public administration units is indicated in the Theoretical Overview part, not earlier. I believe that this issue was not adequately highlighted in the Introduction, which focuses on emphasizing the importance of the prioritization process. Perhaps it has to do with the fact that- according to the authors: “the prioritization process is a good occasion to identify new coordination needs associated to SDG implementation and establish a diagnosis regarding possible levers and barriers to change”. Moreover, at this point it should be emphasized the "transition" in the theoretical part to the issue of SDG - which in my opinion is not a specific program – is incomprehensible and could easily be replaced by any other program implemented in the Ministry under study.
In this context, I believe that Theoretical Overview does not provide an answer to the question of what research problem was discussed in the article and whether it is really related to SD. In my opinion, the theoretical background is also not sufficient. I would like to emphasize that the Change Management Model is widely known, so there is no need to include it (in the form of a drawing) in the text of the article.
The aim of the study is again - this time directly - indicated in 3.1. Research aim as: „paper aims at shedding some light on the way public sector organizations can start the implementation of the SDGs.” - and, as I wrote at the beginning, I have doubts whether the conducted case study is enough for such a broad generalization. Also in 3.1. a research question was indicated, which - in my opinion - is insufficiently connected with the statements contained in the Theoretical Overview. Therefore, I identify the lack of embedding of the research problem in the literature on the subject.
While referring to the results of the research carried out by the Authors, I see here only a descriptive results. SDG goals were described, the organizational units participating in the realization of these goals were presented, their involvement was described and it was pointed which of them participated in the organized workshops. Unfortunately, the interpretations of the results are - in my opinion - not fully valid, as the research methodology is not known, apart from the general indication of the case study methodology. It is known that the research was conducted among the employees of (?) The organizational units of the ministry, but it is not known how the research process looked like, how many people and who participated in the research. And:
- conclusion: „The high participation rate in the thematic workshops was interpreted by the research team as a sign of the disposition of the different organizational units to take part in the process and become involved in collaborative activities related to the SDGs.” – it is not known on what basis such a conclusion was formulated - what variables were tested, and how many participants were there.
- conclusions: „the high participation in the thematic workshops is also important to create a collective readiness of change promoting group sensemaking and shared belief towards change, which are important elements in developing a multilevel perspective of change [54]”. – as before - in my opinion, such inference requires at least a thorough description of the research carried out in the described organization.
It should also be noted that the authors partially refer to the results of the research carried out: „Figure 6 summarizes the main drivers and barriers for collaboration detected as well as the number of times they were mentioned by the participants in the workshops (Recurrence)”. However, these are not sufficiently described in the article presented for review.
Reviewer 4 Report
The paper is presenting the case study of SDG implementation process in the public sector in Spain. The paper has a very nice flow, is easy to read and is written in well english. Overall, enojoyed the reading of it.
Here, I would like to share my comments and suggestions for improvement of the paper.
Introduction section should present bacground, research gap, motivation as well as also the research question, which will be addressed in the paper. In the case of this paper, RQ is presented under Research approach.
What is the importance of figure 1 presented in theoretical overview chapter? How does it relate to your further analyses?
Research methodology - it would be nice, if you could present the diagram or a process of the overall methodology used in case study analysis. You mentioned design science approach - but there is no reference to Hevner et al.
Figure 5 would need more information to be easier understood by readers.
What is the main result of your case study? I assume that this is Figure 6. Figure 6 could be improved. Recurrence is related to what exactly? If you provide figure, it should be clear enough to the reader. Why are forces for change not connected with change proposal with perhaps + sign and forces against change wiht - sign? Please try to improve the figure as this is the main artefact - result of your research.
Please revisit your research question and clearly refer to it, how your research provided to answer to this question.
The results should be presented under Result chapter. Discussion should offer discussion related to previous research results.
Also it would be good to provide a kind of recommended process of change, perhaps like a roadmap, which could be usable also for other public enterprises.
Round 2
Reviewer 3 Report
Thank you for your response to the review and for your detailed and clear response to the comments I sent. The Authors made the corrections that were necessary.